# Does Body Position Influence Bioelectrical Impedance? An Observational Pilot Study

**DOI:** 10.3390/ijerph19169908

**Published:** 2022-08-11

**Authors:** Paweł Więch, Filip Wołoszyn, Patrycja Trojnar, Mateusz Skórka, Dariusz Bazaliński

**Affiliations:** 1Department of Nursing and Public Health, Institute of Health Sciences, College of Medical Sciences, University of Rzeszow, 35-959 Rzeszow, Poland; 2Department of Nursing, Institute of Social Sciences and Health Protection, East European State Higher School in Przemysl, 37-700 Przemysl, Poland; 3Department of Human Physiology, Institute of Medical Sciences, College of Medical Sciences, University of Rzeszow, 35-959 Rzeszow, Poland; 4Medical College, University of Information Technology and Management, 35-225 Rzeszow, Poland; 5Orthopedics Department, St. Hedvig Clinical Provincial Hospital, 35-301 Rzeszow, Poland

**Keywords:** BIA, body composition, impedance, phase angle, nutritional status, anthropometry

## Abstract

As the availability of various bioelectric impedance analysis (BIA) tools is increasing, the patient’s position during the test may be of significant importance for the comparability of the results. An observational pilot study was undertaken between March and May 2021 at the Center for Innovative Research in Medical and Natural Sciences at the University of Rzeszow, Rzeszów, Poland. All participants (*n* = 49: M: 21.05 y ± 1.12 vs. F: 21.34 y ± 2.06) were subjected to measurements of selected nutritional status indicators and body components in three positions: lying, sitting and standing. The body composition indicators were obtained using a bioelectrical impedance device, AKERN BIA 101 Anniversary Sport Edition Analyzer (Akern SRL, Pontassieve, Florence, Italy). The results were analyzed using dedicated software (BodygramPlus 1.2.2.12 from AKERN 2016, Florence, Italy). Our observations indicate that there is a significant difference between lying and standing as well as sitting and standing with respect to anthropometric and nutritional indicators (resistance, reactance, phase angle, standardized phase angle, body cell mass index and fat-free mass index) and body composition components, with particular reference to intracellular and extracellular water. The described differences are significant for both sexes. This study showed that this significantly influenced the scores of components directly related to resistance, reactance and hydrated cell mass, while not affecting the percentages or absolute values of fat and fat-free mass.

## 1. Introduction

The bioelectrical impedance (BIA) method has been widely used for many years in the screening of healthy people [1,2,3] and in-depth investigation of chronically ill people [2,4,5,6,7]. Considered by many to be the gold standard of nutritional status assessment, it gave rise to improved insight into the human body, with a small number of potential contraindications to its use [8,9,10,11]. Its development led to the creation of new measurement tools and their adaptation to the current condition of a patient. BIA is based on the different impedance of fat and lean tissues when a weak electric current flows through the body and several technologies have been designed and commercialized to date [12]. According to recommendations, the devices may either provide the quantitative estimation of body composition parameters using predictive equations set by the manufacturer or provide the raw resistance (R) and reactance (Xc) to be inserted into specific formulas up to the operator [13].

Currently, the most common measurement devices are based on two-electrode configurations (foot–foot or hand–hand) or four-electrode configurations (hand–foot). The accuracy of the measurement depends on several factors, including the reliability of the measurement of the body weight and height, the patient’s position during the test, time since the last meal, the patient’s hydration level, the temperature of the patient and the room in which the test takes place, as well as the time of day of the test [13,14,15,16]. The detailed recommendations for the clinical application of BIA were presented by Kyle et al. [17]. The position of the patient’s body during the test is of great importance. Some researchers have emphasized the advantage of lying over standing [18,19,20] due to the better equalization of fluid levels, and thus more reliable research results, especially in relation to the hydrated spaces of the body. In addition, using the BIA apparatus adapted to the supine position, we were also able to estimate selected body composition parameters in seriously ill people who are immobilized in bed [21,22].

Considering the current state of knowledge, opinions differ. A reliable consensus in the described range is to follow the recommendations of the manufacturer of a given measuring device, with the simultaneous acceptance of the results obtained. However, the main problem is the comparison of the data obtained with a given measuring tool with the results of other researchers who also use the BIA method, but with a different starting position of the patient. There are single research results analyzing different measuring devices in the same study group [18,23,24]. An additional problem is the technological differences that have been reported to potentially correspond to the disagreement between devices, where different frequencies are used in different devices [25]. Low frequency (e.g., 5 kHz) can only provide information on the extracellular water, since the cell membrane cannot be penetrated [13,26]. Nevertheless, poor reproducibility has been noticed at frequencies below 5 kHz and above 200 kHz [25]. To overcome such issues, an intermediate frequency of 50 kHz was proposed as the best sampling frequency. To date, the foot-to-hand technology at 50 kHz single frequency (which was used in our study) is considered the reference method for BIA in humans [13,27,28]. According to the new systematic review presented by Campa et al. [12], good agreement with the reference methods was observed when estimating FFM using predictive equations developed for foot-to-hand technology. However, the generalized equations lead to an underestimation of FFM.

Our study addresses the potential coexistence of measurement differences within the same group and using the same device, but in three different positions: lying (recommended by the company), sitting and standing. In our opinion, the presented results shed new light on an important methodological aspect related to the measurement position recommended by the manufacturer of the device, and the body position of the patient, related to the actual health condition. For this reason, the aim of the study was to thoroughly assess the effect of patient positioning during the BIA test on the reliability of the results obtained for individual body composition components.

## 2. Materials and Methods

### 2.1. Ethics

This study was approved by the institutional Bioethics Committee at the University of Rzeszow (Resolution No. 4/03/2019) and by all appropriate administrative bodies. The study was conducted in accordance with the ethical standards laid down in an appropriate version of the Declaration of Helsinki (64th WMA General Assembly, Fortaleza, Brazil, October 2013) and Polish national regulations.

### 2.2. Subjects

The present observational pilot study was conducted between March and May 2021 at the Center for Innovative Research in Medical and Natural Sciences of the University of Rzeszow among medical students. The study involved a group of 49 students (M: 21.05 y ± 1.12 vs. F: 21.34 y ± 2.06) of the Medical College of Rzeszow University in Rzeszow (Poland). The inclusion criteria were as follows: aged 20 to 30 years, written consent to participate in the study and lack of chronic disease that may affect the physiological and nutritional status, and lack of contraindications specified by the company producing the bioimpedance (BIA) analyzer. Participants who were unwilling or unable to give informed consent or participating in another research project were not accepted. All students invited to the study met the inclusion criteria. The purposes and procedures of the study were explained, and informed consent was obtained. The personal data of subjects were protected by assigning each participant a code in the form of a digital number.

### 2.3. Assessments

Before the main procedure, the body weight and height of the students were measured. The measurements were performed under standard conditions, in an upright position, barefoot, and in a fasting state. Body weight and height were assessed with an accuracy of 0.1 kg/0.1 cm using a digital scale (Radwag 100/200 OW, Radom, Poland). The nutritional indicators were calculated as follows: body mass index (BMI) was calculated as weight (kg)/height (m^2^) (kg/m^2^); body cell mass index (BCMI): BCM (kg)/height (m^2^) (kg/m^2^); skeletal muscle index (SMI): SM (kg)/height (m^2^); fat mass index (FMI): FM (kg)/height (m^2^) (kg/m^2^); fat-free mass index (FFMI): FFM (kg)/height (m^2^) (kg/m^2^). The PA was calculated using the following formula: PA = tangent arc (Xc/R) × 180/π [29].

The AKERN BIA 101 Anniversary Sport Edition Analyzer (Akern SRL, Pontassieve, Florence, Italy) was used to evaluate the differences in the basic components of body composition. The results were analyzed using dedicated software (BodygramPlus 1.2.2.12 from AKERN 2016, Florence, Italy). The equations used by the software to assess the specific parameters are restricted property of the company, but to a significant degree, they are based on computed algorithms developed by Sun S. et al. [30]

In order to assess the reliability of the obtained results of individual components of the body composition in different positions, the same measuring tool, the study group and the test time were used. The whole-body BIA device involved a tetrapolar method. After turning on the device, sinusoidal current with an amplitude of 800 mA and 50 kHz (imperceptible to the human body) passed through the body of the examined person and was then intercepted by the device, giving the result of tissue resistance (resistance (Rz) and reactance (Xc)). The measurements were performed between 7:00 a.m. and noon, on an empty stomach, in the supine position, with abducted upper (30°) and lower (45°) limbs, following at least a 5 min rest. To ensure the fact that the results were reliable and reproducible, two measurements were performed, one after another. Disposable electrodes were placed on the dorsal surface of the right arm (above the wrist) and the right leg (on the ankle). All measurements were performed according to guidelines described by other authors [17,31,32,33].

BIA analysis included: fat mass (FM) (kg and %), fat-free mass (FFM) (kg and %), muscle mass (MM) (kg and %), total body water (TBW) (L and %), intra- and extracellular water (ICW and ECW) (%), body cell mass (BCM) (kg), skeletal muscle mass (SMM) (kg), appendicular skeletal muscle mass (ASSM) (kg) and standardized phase angle (SPA) (°).

### 2.4. Statistical Analysis

The statistical analysis was conducted with Statistica 13.1 (StatSoft Inc., Krakow, Poland). The Shapiro–Wilk test was used to verify the equivalence of the studied groups and the compliance of the distribution of variables with a normal distribution. The ANOVA one-way analysis of variance test was used to assess the impact of body position on the values of nutritional status indicators and body components in women and men. The applied post hoc test was Tukey’s test. Variables are presented as a mean value with standard deviation. The prevalence was calculated with a 95% confidence interval. A *p*-value below 0.05 was considered to be statistically significant.

## 3. Results

Table 1 presents the basic anthropometric, impedance and nutritional parameters among the male participants. There were no significant differences between the body positions (supine, sitting and standing) in any characteristics, except for the phase angle (PA) parameter (F = 4.32; *p* = 0.018). The post hoc test confirmed a significant difference between the PA value in the sitting and standing positions (*p* = 0.020), and the difference in the supine and standing positions was close to the significance threshold (*p* = 0.072). The value of the PA was the highest in the sitting position (7.23 ± 1.40), while the lowest was in the standing position (6.27 ± 0.68). (Table 1).

Among women, the difference between the measurement value of the resistance (F = 4.34; *p* = 0.016) in the sitting and standing positions was significant in the post hoc test (*p* = 0.017). Likewise, for the FFMI index (F = 3.43; *p* = 0.035), a statistically significant difference was found between the sitting and standing positions (*p* = 0.035). In both cases, standing measurement values were the lowest for Xc (61.07 ± 9.11) and FFMI (15.55 ± 0.92), and sitting measurement values were highest for Xc (70.38 ± 14.40) and FFMI (16.21 ± 1.07). We also found significant differences between PA (F = 6.26; *p* = 0.003), BMR (F = 8.89; *p* = 0.002) and BCMI (F = 7.33; *p* = 0.001). For each of these parameters, the values of measurements in the standing position were the lowest relative to the others (PA: 5.40 ± 0.72, Mbasale: 1362.62 ± 62.59 and BCMI: 7.85 ± 0.90) (Table 2).

We also found significant differences for both genders in the analysis of selected body composition parameters. Among men, statistically significant differences were observed in ECW% (F = 4.48; *p* = 0.016), ICW% (F = 4.47; *p* = 0.016) and SPA (F = 4.21; *p* = 0.022). In the post hoc test, significant differences were found for each of the three parameters between the measurement taken in the standing and supine positions and between the standing and sitting positions. The obtained ECW% results in the standing position were the highest (44.63 ± 2.93). In the case of ICW%, the results were lowest when standing (55.37 ± 2.93), while for SPA, the mean results were negative (−0.77 ± 0.89) (Table 3).

Among women, statistically significant differences were found for ECW% (F = 7.22; *p* = 0.001), ICW% (F = 7.22; *p* = 0.001), BCM (F = 6.89; *p* = 0.002) and SPA (F = 6.14; *p* = 0.003). The values of ECW% were highest in the standing position (48.90 ± 3.88), while the values of the BCM, ICW% and SPA measurements were the lowest in the standing position (21.13 ± 2.15, 51.10 ± 3.88 and 0.34 ± 1.21, respectively) (Table 4).

## 4. Discussion

The presented studies analyzed the relationship between the patient’s position during the BIA test and the obtained results for selected nutritional status indicators. Additionally, they focused on the coexisting differences in the components of fat and fat-free mass. Our study is a contribution to the discussion on the credibility of individual measurement tools and their reliability. In our opinion, one of the most important aspects of the methodological correctness of the test, which is the basis for result reliability, is the position in which the patient has measurement analyses performed. Manufacturers of individual types of BIA devices indicate the uniqueness of their products, which is often reflected in the specific restrictions of a given measurement. One of the arguments for this is the claim that there is a difference in the distribution of bodily fluids between the supine and standing positions, which may affect the results of individual components of the body.

American scientists from Texas Tech University [23] analyzed three bioimpedance devices—supine bioimpedance spectroscopy (BIS), supine single-frequency bioelectrical impedance analysis (SFBIA) and standing multi-frequency bioelectrical impedance analysis (MFBIA)—noting the significant differences in Rz, Xc and PA observed between all analyzers. Additionally, the results indicated that Rz and PA values detected by the supine analyzers were equivalent to each other, but not with the standing analyzer. Only the reactance value was comparable between all devices. The study by Tinsley et al. among adult females assessed via multifrequency BIA and bioimpedance spectroscopy indicated that, despite notable differences in the characteristics of the bioimpedance devices and cross-sectional disagreement, strong group-level agreement for detecting changes in R, Xc and PA was generally observed. Notwithstanding, some errors were observed at the individual level [34].

The analysis of the given parameters is extremely important because it is the result of further mathematical transformations of individual composition parameters calculated with the use of a given software. This is especially true for resistance (the opposition offered by the body to the flow of an alternating electrical current, which is inversely related to the water and electrolyte content of the tissue) and reactance (related to the capacitance properties of the cell membrane, with variations occurring depending on its integrity, function and composition). The obtained observations (in part related to Rz, Xc and PA) were only confirmed in our study for PA. Our study demonstrated significant differences between measurements taken in the sitting position and standing position (*p* = 0.020) for men and the lying position vs. sitting (*p* = 0.022) and sitting vs. standing (*p* = 0.004). The value of PA was the highest in the sitting position and the lowest in the standing position for both sexes.

The described observations demonstrate the variability of the PA index depending on the adopted body position. This index is one of the new and increasingly frequently used indicators of nutritional status in health and disease, and is used by many clinicians [35,36,37,38,39]. This also better reflects the degree of cellular health and indicates an early water shift from the intracellular to extracellular compartment in subjects with malnutrition [40,41,42]. In our study, these parameters (ICW and ECW) significantly differed between the lying and standing positions as well as sitting and standing positions, in both women and men. A study by Gibson et al. evaluated the change in hydrated tissue due to changes in body position [20]. They demonstrated a rapid stabilization of TBW, with a simultaneous and longer (over 30 min) stabilization of ICW and ECW.

In addition, significant differences were also observed in the results of measurements of the PA transformation to SPA, calculated by subtracting the reference PA value (dependent on age and sex) from the measured PA and then dividing the result obtained by the reference standard deviation [43]. The SPA parameter is clinically significant because it is associated with a number of nutrition status parameters, and it has been shown to predict postoperative infectious complications [44]. The obtained significant differences in the measured SPA in different positions in our study clearly confirm its dependence on the standard PA, which reflects the cell membrane integrity and BCM [45]. The PA is proportional to the BCM and thus depends on the type of disease affecting the magnitude of the cell membrane potentials [40,46]. Moreover, in our study, the measurement sensitivity of the BCM and its transformation into the BCMI index was demonstrated. The described result confirms the entire cause-and-effect relationship between Rz, Xc, ICW, ECW, PA, SPA and BCM.

Our study is not free from limitations. Despite our efforts to include as many participants as possible, the relatively low number of study participants is the most important limitation. Therefore, the obtained results qualified as a pilot for study I in the search for an empiric response in a larger group of people. Moreover, obtaining a similar age group of young healthy people and their division according to gender should significantly reduce the risk of estimation error. The second limitation of this study was the use of the BIA measuring tool in a manner different from that initially assumed by the manufacturer of the device (in addition to the recommended lying position, measurements were taken in a sitting and standing position). This decision resulted from an attempt to minimize the measurement and interpretation error that, in our opinion, could arise with the use of different measuring devices. We believe that the presented pilot study is a prelude to a deeper discussion of the limitations of BIA measurement and the comparison of the obtained results.

## 5. Conclusions

The adopted position of the body during BIA testing has a significant impact on the measurements of resistance and reactivity, reflected in the intracellular and extracellular hydrated tissue and its cell mass. Moreover, the results of the phase angle, standard phase angle and cellular index significantly differ in function of whether the patient is in the prone, sitting or standing position. However, body position does not affect the percentages and absolute values of overall fat and fat-free mass.

## Figures and Tables

**Table 1 ijerph-19-09908-t001:** Anthropometric and nutritional indicators among men.

Parameters	Man						One-Way ANOVA F/*p*-Value		Post hoc (Tukey’s) *p*-Value		
	L		S		ST						
	Mean	SD	Mean	SD	Mean	SD			L-S	L-ST	S-ST
Height (cm)	180.24	7.90	179.60	7.53	179.95	7.57	0.04	0.965	-	-	-
Age (years)	21.05	1.12	21.05	1.15	21.00	1.15	0.01	0.988	-	-	-
Weight (kg)	74.68	11.18	73.90	10.86	74.05	11.13	0.03	0.971	-	-	-
Rz (ohm)	535.52	75.63	534.40	52.55	554.63	56.33	0.64	0.531	-	-	-
Xc (ohm)	65.95	12.36	67.55	14.17	61.05	9.51	1.49	0.234	-	-	-
PA (°)	7.03	0.99	7.23	1.40	6.27	0.68	4.32	**0.018**	0.833	0.072	**0.020**
BMR (kcal)	1767.40	166.57	1769.94	153.43	1671.55	109.20	2.87	0.065	-	-	-
BMI (kg/m²)	23.00	3.12	22.93	3.18	22.89	3.26	0.01	0.994	-	-	-
BCMI (kg/m²)	10.77	1.50	10.93	1.67	9.83	1.19	3.17	0.050	-	-	-
SMI (kg/m²)	9.94	1.09	9.89	0.78	9.61	0.83	0.75	0.479	-	-	-
FMI (kg/m²)	4.53	2.16	4.41	2.10	5.01	1.88	0.46	0.635	-	-	-
FFMI (kg/m²)	18.47	1.88	18.52	1.71	17.87	1.73	0.81	0.451	-	-	-

Abbreviations: L—supine position; S—sitting position; ST—standing position; Rz—resistance; Xc—reactance; PA—phase angle; BMR—basal metabolic rate; BMI—body mass index; BCMI—body cell mass index; SMI—skeletal muscle index; FMI—fat mass index; FFMI—fat-free mass index. Bold characters indicate significant values (*p* < 0.05).

**Table 2 ijerph-19-09908-t002:** Anthropometric and nutritional indicators among women.

Parameters	Women						One-Way ANOVA F/*p*-Value		Post hoc (Tukey’s) *p*-Value		
	L		S		ST						
	Mean	SD	Mean	SD	Mean	SD			L-S	L-ST	S-ST
Height (cm)	164.34	5.42	164.34	5.42	164.34	5.42	0.00	1.000	-	-	-
Age (years)	21.34	2.06	21.34	2.06	21.34	2.06	0.00	1.000	-	-	-
Weight (kg)	56.83	6.18	56.83	6.18	56.83	6.18	0.00	1.000	-	-	-
Rz (ohm)	619.86	64.64	617.24	52.15	646.14	49.46	2.38	0.099	-	-	-
Xc (ohm)	68.31	13.76	70.38	14.40	61.07	9.11	4.34	**0.016**	0.808	0.080	**0.017**
PA (°)	6.35	1.55	6.56	1.54	5.40	0.72	6.26	**0.003**	0.814	**0.022**	**0.004**
BMR (kcal)	1439.71	113.39	1460.78	130.16	1362.62	62.59	6.89	**0.002**	0.730	**0.019**	**0.002**
BMI (kg/m²)	21.04	2.04	21.04	2.04	21.04	2.04	0.00	1.000	-	-	-
BCMI (kg/m²)	8.80	1.31	9.06	1.53	7.85	0.90	7.33	**0.001**	0.720	**0.015**	**0.001**
SMI (kg/m²)	7.87	0.69	7.87	0.53	7.58	0.52	2.34	0.102	-	-	-
FMI (kg/m²)	5.01	1.78	4.83	1.71	5.48	1.49	1.17	0.315	-	-	-
FFMI (kg/m²)	16.03	1.00	16.21	1.07	15.55	0.92	3.43	**0.037**	0.781	0.157	**0.035**

Abbreviations: L—supine position; S—sitting position; ST—standing position; Rz—resistance; Xc—reactance; PA—phase angle; BMR—basal metabolic rate; BMI—body mass index; BCMI—body cell mass index; SMI—skeletal muscle index; FMI—fat mass index; FFMI—fat-free mass index. Bold characters indicate significant values (*p* < 0.05).

**Table 3 ijerph-19-09908-t003:** Selected components of body composition among men.

Parameters	Male						One-Way ANOVA F/*p*-Value		Post hoc (Tukey’s) *p*-Value		
	L		S		ST						
	Mean	SD	Mean	SD	Mean	SD			L-S	L-ST	S-ST
FFM (kg)	60.06	8.01	59.74	6.29	57.83	6.26	0.59	0.556	-	-	-
FFM (%)	80.95	7.45	81.50	6.72	78.71	5.36	0.97	0.385	-	-	-
TBW (L)	43.48	6.29	42.92	4.80	42.13	4.87	0.31	0.731	-	-	-
TBW (%)	58.60	6.22	58.51	4.45	57.29	4.13	0.40	0.669	-	-	-
ECW (%)	41.65	3.62	41.26	4.72	44.63	2.93	4.48	**0.016**	0.944	**0.045**	**0.022**
BCM (kg)	35.09	5.75	35.18	5.29	31.77	3.77	2.89	0.064	-	-	-
FM (kg)	14.62	7.03	14.16	6.83	16.23	6.14	0.51	0.603	-	-	-
FM (%)	19.05	7.45	18.51	6.72	21.29	5.36	0.97	0.385	-	-	-
ICW (%)	58.35	3.62	58.74	4.72	55.37	2.93	4.47	**0.016**	0.944	**0.045**	**0.022**
MM (kg)	32.37	4.82	31.90	3.14	31.11	3.03	0.56	0.574	-	-	-
MM (%)	43.82	6.44	43.66	4.76	42.51	4.46	0.35	0.705	-	-	-
SMM (kg)	32.37	4.82	31.90	3.14	31.11	3.03	0.56	0.574	-	-	-
ASMM (kg)	24.96	3.54	24.67	2.63	23.78	2.59	0.83	0.438	-	-	-
SPA (°)	0.25	1.31	0.50	1.92	−0.77	0.89	4.21	**0.020**	0.847	0.074	**0.022**

Abbreviations: L—supine position; S—sitting position; ST—standing position; FFM—fat-free mass; TBW—total body water; ECW—extracellular water; BCM—body cell mass; FM—fat mass; ICW—intracellular water; MM—muscle mass; SMM—skeletal muscle mass; ASMM—appendicular skeletal muscle mass; SPA—standardized phase angle. Bold characters indicate significant values (*p* < 0.05).

**Table 4 ijerph-19-09908-t004:** Selected components of body composition among women.

Parameters	Female						One-Way ANOVA F/*p*-Value		Post hoc (Tukey’s) *p*-Value		
	L		S		ST						
	Mean	SD	Mean	SD	Mean	SD			L-S	L-ST	S-ST
FFM (kg)	43.30	3.38	43.77	3.52	41.98	2.60	2.44	0.093	-	-	-
FFM (%)	76.70	7.17	77.52	6.89	74.32	5.09	1.93	0.152	-	-	-
TBW (L)	31.45	2.76	31.47	2.50	30.66	2.03	1.04	0.357	-	-	-
TBW (%)	55.76	5.89	55.74	4.91	54.27	3.85	0.87	0.423	-	-	-
ECW (%)	44.89	5.60	44.06	5.84	48.90	3.88	7.22	**0.001**	0.813	**0.012**	**0.002**
BCM (kg)	23.79	3.91	24.51	4.49	21.13	2.15	6.89	**0.002**	0.729	**0.019**	**0.002**
FM (kg)	13.53	4.99	13.06	4.83	14.85	4.22	1.13	0.328	-	-	-
FM (%)	23.30	7.17	22.48	6.89	25.68	5.09	1.93	0.152	-	-	-
ICW (%)	55.11	5.60	55.94	5.84	51.10	3.88	7.22	**0.001**	0.813	**0.012**	**0.002**
MM (kg)	21.31	2.44	21.29	2.05	20.43	1.42	1.77	0.176	-	-	-
MM (%)	37.81	5.47	37.75	4.37	36.23	3.21	1.18	0.312	-	-	-
SMM (kg)	21.31	2.44	21.29	2.05	20.43	1.42	1.77	0.176	-	-	-
ASMM (kg)	16.69	1.99	16.80	1.98	15.69	1.28	3.38	**0.039**	0.969	0.091	0.053
SPA(°)	1.92	2.58	2.26	2.58	0.34	1.21	6.14	**0.003**	0.828	**0.023**	**0.004**

Abbreviations: L—supine position; S—sitting position; ST—standing position; FFM—fat-free mass; TBW—total body water; ECW—extracellular water; BCM—body cell mass; FM—fat mass; ICW—intracellular water; MM—muscle mass; SMM—skeletal muscle mass; ASMM—appendicular skeletal muscle mass; SPA—standardized phase angle. Bold characters indicate significant values (*p* < 0.05).

## Data Availability

The data presented in this study are available upon reasonable request from the corresponding author: pwiech@ur.edu.pl.

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
