# Peer review of "Does Body Position Influence Bioelectrical Impedance? An Observational Pilot Study"

_ijerph, 2022, doi:10.3390/ijerph19169908_

Round 1

Reviewer 1 Report

The manuscript presents a couple of major points that must be addressed before a possible recommendation for acceptance.

First, there is no rationale to justify the calculation of the BIA-derived parameters (e.g., FM, FFM, SMM …). In this regard, the prediction equations used are not reported by the authors in the text, therefore it is not possible correctly understand the mechanisms underlying the current results. Therefore, my request is to maintain only the analysis performed on the raw parameters and to completely remove those on the BIA-derived derived values.

Second, the introduction does not provide a comprehensive exploration of the topic.I would encourage to authors to read more about the subject as these aspects need to be considered in the introduction to form the basis for this paper:

- Campa et al.,  Eur J Appl Physiol. 2022 Mar;122(3):561-589. doi: 10.1007/s00421-021-04879-y.

Round 2

Response to Reviewer 1 Comments

Dear Reviewer,

Thank you for your constructive comments. We have revised the manuscript in accordance to your guidelines. All changes have been highlighted in red in the revised version of the manuscript. In the following pages are our point-by-point responses to each of the comments.

General comments to the Authors

  1. First, there is no rationale to justify the calculation of the BIA-derived parameters (e.g., FM, FFM, SMM …). In this regard, the prediction equations used are not reported by the authors in the text, therefore it is not possible correctly understand the mechanisms underlying the current results. Therefore, my request is to maintain only the analysis performed on the raw parameters and to completely remove those on the BIA-derived derived values.

Re: Thank you for your valuable remark. In line with the recommendations, the methodological part indicates (in red) the method of estimating the directional indicators and nutritional indexes (RX, XC, BMI, BCMI etc.). As we mentioned, the equations used by the software to assess the specific parameters are restricted property of the company, but to a significant degree, they are based on computed algorithms developed by Sun S. et al. (Sun et al. 2003 doi: 10.1093 / ajcn / 77.2.331). - we also added this information in the methodological part.

Due to the fact that it is not possible to obtain exact formulas from the company, we have indicated the exact name of the software (BodygramPlus 1.2.2.12 from AKERN 2016, Italy), which is currently the most commonly used by researchers using the most common  hand to foot method of BIA (as mentioned in Campa 2022 in the review literature).

Due to our experience with the use of the device directly in to the patient, and the desire to quickly exchange information (data) between clinicians from different countries, we ask for consent to leave the calculated values ​​of the body composition (FM, FFM and its derivatives), with the method, software (with company-owned designs) and the frequency precisely described.

  1. Second, the introduction does not provide a comprehensive exploration of the topic.I would encourage to authors to read more about the subject as these aspects need to be considered in the introduction to form the basis for this paper: - Campa et al., Eur J Appl Physiol. 2022 Mar;122(3):561-589. doi: 10.1007/s00421-021-04879-y

Re: Thank you for your comment. We have read the proposed publication in detail and we have supplemented our manuscript with it (in the introductory part, in red). Additionally, our introduction is largely based on Campa et al. publications from previous years.

Reviewer 2 Report

The authors have addressed my concerns. Thank you.

Round 2

Response to Reviewer 2 Comments

General comments to the Authors

  1. The authors have addressed my concerns. Thank you.

Re: Thank you for your valuable comments